# Genome-Wide Identification and Expression Analysis of the *ClHMGB* Gene Family in Watermelon Under Abiotic Stress and *Fusarium oxysporum* Infection

**DOI:** 10.3390/ijms27010157

**Published:** 2025-12-23

**Authors:** Changqing Xuan, Mengli Yang, Yufan Ma, Xue Dai, Shen Liang, Gaozheng Chang, Xian Zhang

**Affiliations:** 1Institute of Horticultural Research, Henan Academy of Agricultural Sciences, Zhengzhou 450002, China; 19829661028@139.com (C.X.); yang857215426@163.com (M.Y.); myf18595726255@163.com (Y.M.); liangshen1971@hnagri.org.cn (S.L.); 2State Key Laboratory of Crop Stress Biology in Arid Areas, College of Horticulture, Northwest A&F University, Xianyang 712100, China; dai_xue@163.com

**Keywords:** HMGB, watermelon, genome-wide identification, expression pattern, abiotic stresses, *Fusarium oxysporum*

## Abstract

High-Mobility Group B (HMGB) proteins are conserved non-histone nuclear proteins involved in DNA replication, transcription, recombination, repair; plant growth and development; and stress responses. In this study, we identified nine *ClHMGB* genes in watermelon using genome-wide search. Phylogenetic and homology analyses classified them into four distinct classes. Synteny analysis revealed that *ClHMGB* genes share closer evolutionary relationships with dicots than with monocots. Tissue-specific expression profiling showed that eight *ClHMGB* members exhibit higher transcript levels in female and/or male flowers, suggesting that they play essential roles in floral organ development. Under drought, low-temperature, and salt stresses, *ClHMGB* members displayed distinct expression patterns. For instance, *ClHMGB4* and *ClHMGB8* were downregulated under drought and low-temperature stress but upregulated under salt stress, indicating potential functional specialization in response to different abiotic stresses. The highly virulent *Fusarium oxysporum* f. sp. *niveum* race 2 (*Fon* R2) induced the upregulation of more *ClHMGB* genes than the less virulent race 1 (*Fon* R1). Four members (*ClHMGB1*, *4*, *6*, and *7*) were consistently upregulated by both races, suggesting that they may play fundamental roles in disease resistance. This study provides a foundation for further investigation into the roles of *ClHMGB* genes in growth, development, and stress responses of watermelon.

## 1. Introduction

High-Mobility Group (HMG) proteins are the second most abundant class of nuclear proteins in eukaryotes, after histones [1]. They are named for their rapid mobility during gel electrophoresis, a consequence of their relatively small size [2]. Functionally, HMG proteins act as architectural cofactors for DNA in the nucleus and mitochondria, serve as signaling regulators in the cytoplasm, and can also function as inflammatory cytokines in the extracellular space [3,4,5]. Based on their structural domains, HMG proteins are classified into three major families: the HMGA family, which contains A/T-hook DNA-binding motifs; the HMGN family, characterized by a nucleosome-binding domain and found exclusively in mammals; and the HMGB family, harboring the HMG-box domain, which typically consists of about 75 amino acid residues folded into an L-shaped structure, with the longer arm formed of the N-terminal strand and helix III, and the shorter arm by helices I and II [6,7,8]. In plants, HMG proteins belong to the HMGA and HMGB families. Plant HMGB proteins can be further divided into four subclasses according to their domain organization: the canonical HMGB type (the most abundant subgroup, containing a single HMG-box domain); the 3xHMG-box type (containing three HMG-box domains and an N-terminal basic region); the ARID-HMG type (containing an AT-rich interaction domain (ARID) alongside an HMG-box domain); and the SSRP1 type (containing a structure-specific recognition protein (SSRP1) domain followed by an HMG-box domain) [9,10].

To date, *HMGB* genes have been identified and functionally characterized in a number of plant species, including *Arabidopsis* [11,12], rice [13], maize [14], cucumber [15], grape [16], *Betula platyphylla* [17], pea [18], citrus [19], and tomato [20]. Previous reports have indicated that HMGB proteins are involved in a wide range of critical biological processes, such as the regulation of DNA replication, transcription, recombination, and repair, and also participate in plant growth and development [9,21]. For example, cold stress upregulated the transcript levels of *AtHMGB2*, *AtHMGB3*, and *AtHMGB4* [22], whereas drought and salt stress led to marked downregulation of *AtHMGB2* and *AtHMGB3*. Under salt and drought conditions, transgenic *Arabidopsis* overexpressing *AtHMGB2* showed delayed germination and subsequent growth retardation compared with wild-type plants, while *AtHMGB4* overexpression had no discernible effect. Interestingly, although *AtHMGB5*-overexpressing plants displayed no notable stress-related phenotypes, *hmgb5* mutants exhibited germination and growth under stress [17,22]. In maize, ectopic expression of *ZmHMGB1* inhibited primary root elongation in tobacco seedlings, whereas *ZmHMGB4* had no observable influence on root development [23]. In rice, phosphorus (Pi) deficiency induced upregulation of *OsHMGB1* expression, and the *hmgb1* mutant showed stunted and weak growth under both high- and low-Pi conditions, whereas *OsHMGB1* overexpression increased Pi accumulation in leaves and roots. Further studies revealed that OsHMGB1 maintains Pi homeostasis by binding to promoters of phosphate starvation response (PSR) genes and regulating their expression [24]. In citrus, the *FhARID1* gene, which belongs to the ARID-HMG class, was found to regulate axillary bud proliferation: *FhARID1* knockout led to aberrant bud proliferation and a compact plant phenotype, while its overexpression produced no discernible change compared with wild-type plants [19]. In *Arabidopsis*, the *SSRP1* gene has been implicated in seed dormancy, with *ssrp1* mutants exhibiting enhanced germination rates in freshly harvested seeds and a shortened dormancy period [25]. In cucumber, the *FS2* gene encodes an ARID-HMG transcription factor that positively regulates fruit spine density by activating *Tril* expression. Consequently, *fs2* mutants showed fewer spines on fruits and fewer trichomes on leaf surfaces [26]. Beyond these specific roles, the HMGB protein has also been shown to participate in root development [23], leaf growth [27], flowering induction [28], flower development [29], and cell division.

In addition, some plant *HMGB* genes have been demonstrated to participate in responses to abiotic and biotic stresses. For example, in *Larix principis-rupprechtii*, *LpHmgB10* is involved in reactive oxygen species (ROS) metabolism and signaling, acting as a key downstream regulator of the transcription factor *LpWRKY64.* Both *LpWRKY65* and *LpHmgB10* overexpression enhanced ROS scavenging, which helped maintain redox homeostasis and thereby improved the embryogenic capacity of *Larix principis-rupprechtii* [30]. Lei et al. identified 11 *HMGB* genes in *Betula platyphylla*, and among them, *BpHMGB6* was markedly upregulated under salt stress. *BpHMGB6* overexpression increased the activity of protective enzymes and strengthened ROS-scavenging capacity, reducing salt-induced cellular damage and death. In contrast, transient silencing of this gene resulted in the opposite physiological effects, indicating that it has a role in salt tolerance [17]. In rice, drought and salt stress treatments significantly induced the expression of *OsHMGB707*. Overexpression of this gene notably improved drought tolerance, while its silencing did not reduce drought resistance, likely due to functional compensation by redundant *HMGB* members [31]. Interestingly, *HMGB* genes also play a role in mitigating phytotoxicity. In cucumber, the *CsHMGB* gene was induced by propamocarb. Its overexpression enhanced wax deposition, increased stomatal conductance in the pericarp cells, and activated the antioxidant system, collectively alleviating phytotoxic damage. Conversely, silencing this gene led to greater propamocarb residue accumulation [32]. Research on the involvement of *HMGB* genes in biotic stress responses remains relatively limited. HMGB3 is recognized as a damage-associated molecular pattern (DAMP) in both animals and plants [33,34,35]. For example, in *Arabidopsis*, HMGB3 is released into the apoplast upon *Botrytis cinerea* infection, where it activates innate immune responses via the receptor-like kinases BAK1 and BKK1, thereby enhancing resistance to this pathogen. Silencing *HMGB3* increases susceptibility, whereas exogenous application of the HMGB3 protein restores disease resistance [36]. In rice, OsHMGB1 functions as a negative regulator in the interaction with *Xanthomonas oryzae* pv. *oryzae* (*Xoo*). Both RNAi and knockout lines of *OsHMGB1* conferred stronger resistance to *Xoo* strains, while *OsHMGB1* overexpression increased susceptibility to bacterial blight [37]. Moreover, OsHMGB1 competes with the calmodulin-like protein OsCML31 for the opportunity to bind to the nematode effector MgCRT1. *OsCML31* overexpression and *OsHMGB1* knockout increased the susceptibility of rice to root-knot nematodes, whereas *OsCML31* knockout and *OsHMGB1* overexpression had the opposite effect [38].

Watermelon crops are widely cultivated, and their fruit is popular with consumers on a global scale due to its appealing taste and texture. However, their growth and yield are often compromised by a range of abiotic and biotic stresses, leading to significant reductions in yield and quality. While *HMGB* genes have been implicated in processes such as tissue development and abiotic stress responses in several plant species, their functions in watermelon remain largely unexplored. In this study, nine putative *HMGB* genes were identified in watermelon. A comprehensive analysis was performed, with a focus on their chromosomal locations, protein characteristics, phylogenetic relationships, synteny with other species, *cis*-acting elements in promoter regions, tissue-specific expression patterns, and expression profiles under abiotic and biotic stresses. Together, the findings of this study provide a valuable foundation for elucidating the potential roles of plant HMGB proteins and the specific functions of *ClHMGB* genes in watermelon growth, development, and stress responses.

## 2. Results

### 2.1. Identification and Synteny Analysis of the ClHMGB Genes in Watermelon

In this study, nine putative *HMGB* genes were identified in watermelon by performing a BLAST search of the whole-genome sequences of *Citrullus lanatus* subsp. *vulgaris* cv. 97103 (v2.0) using HMGB protein sequences from *Arabidopsis* as queries. These nine genes, designated as *ClHMGB1* to *ClHMGB9* according to their chromosomal locations, are distributed across seven chromosomes (Figure 1). Chromosomes 1 and 10 each harbor two *ClHMGB* members, whereas chromosomes 4, 5, 6, and 11 contain only one *ClHMGB* gene each. Based on genomic data, the *ClHMGB* genes have genomic sequence lengths ranging from 1073 to 6714 bp, coding sequence (CDS) lengths ranging from 441 to 1929 bp, protein lengths ranging from 147 to 643 amino acids, and molecular weights ranging from 16.11 to 71.61 kDa. Isoelectric point (pI) analysis revealed that ClHMGB1, 3, 5, and 8 are acidic, while ClHMGB2, 4, 6, 7, and 9 are basic (Table 1). In addition, subcellular localization prediction indicated that all ClHMGB proteins are localized in the nucleus. To validate this prediction, subcellular localization experiments were conducted for ClHMGB1 and ClHMGB2. The results demonstrated that these two proteins are localized to the nuclear region of tobacco leaf cells, which is consistent with the prediction (Appendix A).

### 2.2. Phylogenetic and Syntenic Analyses of the ClHMGBs

To examine the evolutionary relationships and potential functions of the *ClHMGB* genes, a comparative phylogenetic and synteny analysis of *HMGB* gene members from watermelon, *Arabidopsis*, rice, and tomato was performed (Figure 2). An unrooted phylogenetic tree was constructed with MEGA 7.0 software, incorporating 9, 11, 11, and 14 HMGB members from watermelon, tomato, rice, and *Arabidopsis*, respectively. Following the established classification of *Arabidopsis HMGB* genes, the tree was divided into four classes: HMGB, ARID-HMG, SSRP1, and 3xHMG-box (Figure 2a). The HMGB clade contained the largest number of HMGB members, including seven from *Arabidopsis*, six from rice, five from tomato, and five from watermelon. The ARID-HMG clade was the second largest, with a total of 12 members: four from *Arabidopsis*, two from rice, four from tomato, and two from watermelon. The SSRP1 and 3xHMG-box clades each contained five members overall, with each clade including only one representative from tomato and one from watermelon, respectively.

A total of six pairwise synteny analyses were performed among the four species. The number of syntenic *HMGB* gene pairs identified in each comparison was as follows: nine between watermelon and *Arabidopsis*, six between watermelon and tomato, three between watermelon and rice, seven between tomato and *Arabidopsis*, two between tomato and rice, and four between rice and *Arabidopsis* (Figure 2b, Appendix A). Consistent with their evolutionary relationships, significantly more syntenic *HMGB* gene pairs were identified between dicotyledon–dicotyledon comparisons than between dicotyledon–monocotyledon comparisons, suggesting early evolutionary divergence of this gene family. Notably, *ClHMGB4* and *ClHMGB9* showed syntenic relationships with all three of the other species examined, implying that these two genes may possess indispensable, conserved functions in plant growth and development.

### 2.3. Gene Structure, Conserved Domains and Cis-Acting Regulatory Elements of the ClHMGBs

The structural conservation and diversity of *ClHMGB* gene members were assessed by analyzing their exon–intron organization within a phylogenetic framework, using full-length coding sequences and corresponding genomic DNA sequences (Figure 3a). The number of exons in these gene members varied from 6 to 16. Specifically, three members (*ClHMGB1*, *3*, and *6*) contained eight exons, while another three (*ClHMGB2*, *4*, and *5*) possessed six exons. Notably, even members in the sister branch exhibited considerable divergence in gene length and structure, revealing substantial evolutionary diversity within this gene family. To further examine conserved domains and motifs in HMGB proteins, we first performed multiple sequence alignment of HMGB protein sequences from the four species. Then, the conserved motifs in ClHMGB were identified using the online tool MEME (Figure 3b and Appendix A). Five distinct motifs were detected among the nine ClHMGB members. Motifs 1 and 2 were universally conserved, as they were present in every member. In contrast, Motifs 3, 4, and 5 were exclusively found in ClHMGB4 and ClHMGB5, both of which belong to the ARID-HMG clade.

To identify *cis*-acting elements in the promoter regions of *ClHMGB* genes, the 2.0 kb sequences upstream of the start codons were retrieved and analyzed using the PlantCARE database. A total of 26 distinct types of *cis*-acting elements were identified and grouped into three functional categories: phytohormone responsiveness, stress responsiveness, and growth and development (Figure 4). The promoters contained eight types of elements responsive to five major hormones: ABRE (abscisic acid); GARE-motif, P-box, and TATC-box (gibberellin); TCA-element (salicylic acid); TGA-element (auxin); and TGACG-motif and CGTCA-motif (jasmonic acid). Five stress-responsive elements were detected: ARE (anaerobic induction), CCAAT-box (heat stress), LTR (low-temperature response), MBS (drought stress), and TC-rich repeats (defense and stress responses). The most abundant category was related to growth and development, which included elements such as CAT-box (meristem expression), circadian element (circadian rhythm), GCN4_motif (endosperm expression), O2-site (zein metabolism regulation), and multiple light-responsive motifs (AE-box, ATCT-motif, Box 4, G-box, GATA-motif, GT1-motif, I-box, MRE, TCCC-motif, and TCT-motif). Although the promoters of all *ClHMGB* members contained *cis*-acting elements related to phytohormone responsiveness, stress responsiveness, and growth and development, those associated with developmental processes—particularly light responsiveness—were particularly abundant. This suggests that *ClHMGB* genes may play important and potentially conserved roles in plant responses to light. In summary, the presence of diverse *cis*-acting elements in *ClHMGB* promoters indicates that these genes may be involved in plant growth and development by responding to various environmental factors.

### 2.4. Profiles of Tissue-Specific Expression Patterns of ClHMGBs

To elucidate the potential functions of *ClHMGB* genes in different tissues, transcript levels were measured by quantitative reverse transcription PCR (qRT-PCR) in samples collected from roots, stems, leaves, tendrils, and female and male flowers. As shown in Figure 5, transcripts of most *ClHMGB* genes were detectable across a wide range of tissues; however, distinct expression profiles were observed among individual members. Specifically, *ClHMGB1* was most abundant in stems, tendrils, and female flowers; *ClHMGB2* was primarily detected in stems, female flowers, and male flowers; *ClHMGB3* and *ClHMGB6* were specific to female flowers; *ClHMGB4* was highly expressed in tendrils and female flowers; *ClHMGB5* was specific to tendrils; *ClHMGB7* was elevated in stems and female flowers; *ClHMGB8* accumulated in tendrils, female flowers, and male flowers; and *ClHMGB9* was highly expressed in stems, tendrils, and both female and male flowers. Notably, all *ClHMGB* gene members displayed relatively high transcript levels in female flower tissues, suggesting that they could play essential roles in female flower development. Collectively, the tissue-specific expression patterns of *ClHMGB* genes imply their broad involvement in multiple aspects of plant growth and development.

### 2.5. Patterns of Expression of ClHMGB Genes in Response to Drought, Low-Temperature, and Salt Stress

Given the presence of stress-responsive *cis*-acting elements in the promoters of *ClHMGB* genes, which suggests their potential induction under adverse conditions, we examined the expression patterns of *ClHMGB* genes in watermelon seedlings subjected to drought, low-temperature, and high-salinity treatments. Under drought stress, most *ClHMGB* members were generally downregulated (Figure 6a). Notably, transcript levels of *ClHMGB4*, *7*, *8*, and *9* were significantly suppressed throughout the stress period (Figure 6b). In contrast, *ClHMGB1* and *ClHMGB2* showed transient upregulation only at 4 days post-treatment (dpt), while *ClHMGB5* and *ClHMGB6* were upregulated at 8 dpt.

During low-temperature stress, four members (*ClHMGB2*, *4*, *5*, and *8*) were consistently downregulated (Figure 6c,d). Conversely, *ClHMGB1*, *7*, and *9* displayed early downregulation (6 and 12 h post-treatment (hpt)) followed by upregulation at later time points (24 and 48 hpt). Meanwhile, *ClHMGB3* predominantly showed a trend of upregulation, with only a slight decrease at 12 hpt. Expression profiles under high-salinity treatment revealed distinct responses (Figure 6e). Throughout the salt stress period, *ClHMGB3*, *4*, and *5* were continuously upregulated (Figure 6f), while *ClHMGB7* exhibited progressive downregulation. *ClHMGB8* and *ClHMGB9* shared a similar expression pattern: both were downregulated at 12 hpt but upregulated at all other time points. *ClHMGB2* was initially downregulated, but later showed substantial transcript accumulation, whereas *ClHMGB6* displayed the opposite trend, with marked downregulation only toward the end of the treatment. Collectively, *ClHMGB* members displayed diverse expression patterns under different abiotic stresses, suggesting that they may carry out distinct biological functions during plant stress adaptation.

### 2.6. Expression Patterns of the ClHMGB Genes in Response to Infection with Fon

To explore the potential roles of *ClHMGB* genes in plant immunity, we analyzed their expression patterns in watermelon seedlings inoculated with *Fusarium oxysporum* f. sp. *niveum* (*Fon*) races 1 and 2 (*Fon* R1, *Fon* R2). As shown in Figure 7a five members (*ClHMGB1*, *3*, *4*, *5*, and *8*) were upregulated at 0 dpt in response to *Fon* R1, whereas eight genes (all except *ClHMGB2*) were upregulated at 0 dpt in response to *Fon* R2. At 3 dpt, four genes (*ClHMGB1*, *4*, *6*, and *7*) were induced by *Fon* R1, and seven genes (*ClHMGB1*, *4–9*) were induced by *Fon* R2. At 7 dpt, four more genes (*ClHMGB3*, *4*, *6*, and *7*) were upregulated under *Fon* R1, while seven genes (*ClHMGB1–7*, and *9*) were upregulated under *Fon* R2.

Notably, at each time point, the more virulent *Fon* R2 strain induced the upregulation of more *ClHMGB* genes than did *Fon* R1. Moreover, all genes induced by *Fon* R1 were also upregulated by *Fon* R2. For example, at 0 dpt, *ClHMGB1*, *3*, *4*, *5*, and *8* were upregulated in response to both races. In addition, *ClHMGB1*, *4*, *6*, and *7* were strongly induced at all three time points by both *Fon* R1 and *Fon* R2, except for *ClHMGB1* at 7 dpt with *Fon* R1, and *ClHMGB6* and *ClHMGB7* at 0 dpt with *Fon* R1 (Figure 7b). Their consistent and pronounced activation suggests that these four genes could serve as key regulators in watermelon disease resistance.

## 3. Discussion

Plant HMGB proteins are a subclass of High-Mobility Group (HMG) proteins defined by the presence of the HMG-box domain. *HMGB* genes have previously been identified in a range of plant species, including *Arabidopsis* [11,12], rice [13], maize [14], cucumber [15], grape [16], *Betula platyphylla* [17], pea [18], citrus [19], and tomato [20]. Moreover, several studies have revealed the involvement of certain *HMGB* genes in processes such as tissue development and abiotic stress responses. However, the functions of the *HMGB* genes in watermelons remain largely unknown. In this study, nine candidate *HMGB* genes in watermelon were identified for the first time through genome-wide screening. A systematic analysis of these genes and their encoded proteins was conducted using molecular biology, genomics, and statistical approaches, with the aim of providing a theoretical basis for elucidating the potential functions of *HMGB* genes in watermelon. The genes, designated *ClHMGB1–ClHMGB9* according to their chromosomal positions, are distributed across all seven watermelon chromosomes (Figure 1). Among them, *ClHMGB2*, *6*, *7*, and *9* are located near chromosome termini, suggesting their potential involvement in chromatin organization and maintenance [39]. Following the classification established for *Arabidopsis* HMGB subfamilies [16,40], the nine *ClHMGB* members were phylogenetically grouped into four clades: HMGB, ARID-HMG, 3xHMG-box, and SSRP1 (Figure 2a). Notably, 3xHMG-box proteins are plant-specific and are encoded in genomes ranging from lower to higher plants. Here, we found that only a single gene encodes a 3xHMG-box protein in each of rice, tomato, and watermelon within this clade, consistent with previous reports [16,17,20,40]. Synteny analysis of *HMGB* genes among watermelon, tomato, *Arabidopsis*, and rice revealed more syntenic gene pairs between watermelon and *Arabidopsis* than in other comparisons. Furthermore, dicotyledon–dicotyledon comparisons yielded more syntenic pairs than dicotyledon–monocotyledon comparisons, indicating that the separation of *HMGB* genes was accompanied by evolutionary species divergence. Additionally, syntenic relationships were also detected for *ClHMGB4* and *ClHMGB9* between watermelon and the other three species, indicating that these orthologs share a relatively ancient ancestral origin. A similar evolutionary pattern has been reported for certain *SlHMGB* members in tomato [20].

HMGB proteins have been implicated in diverse plant developmental processes, including root formation, leaf growth, flowering induction, flower development, and cell division [27,29,41,42]. In *Arabidopsis*, the ARID-HMG DNA-binding protein AtHMGB15 is preferentially expressed in pollen grains and pollen tubes, where it interacts with the transcription factors AGL66 and AGL104 to regulate pollen tube development. Knockdown of *AtHMGB15* results in delayed pollen tube growth and reduced seed set [29]. Additionally, AtHMGB15 forms a transcriptional activation complex with the transcription factor MYC2 to promote the expression of the key jasmonic acid (JA) signaling genes, *MYB21* and *MYB24*. Consequently, the *athmgb15* mutant shows significantly lower levels of JA and JA derivatives, leading to defects in pollen morphology and germination, both of which can be rescued by exogenous methyl jasmonate application [43]. In watermelon, *ClHMGB4*—a homolog of *AtHMGB15*—displayed markedly higher mRNA accumulation in female flowers than in other tissues (Figure 5), suggesting that it could play a regulatory role in female flower development. Moreover, all *ClHMGB* members except for *ClHMGB5* showed elevated transcript levels in male and/or female flowers (Figure 5), suggesting that this gene family may play key regulatory roles in floral organ development. A similar expression pattern has been reported for *SlHMGB* family members in tomato [20]. In contrast, no *ClHMGB* genes were highly expressed specifically in leaf tissues, implying that ClHMGB proteins may perform only basal functions during watermelon leaf development. The 3xHMG-box protein subfamily is unique to plants, with flowering plant genomes encoding one or two members. These proteins are thought to be involved in cell-division-related processes such as chromatin condensation and segregation [44]. Here, a single 3xHMG-box gene, *ClHMGB6*, was identified in watermelon, and its elevated expression in female flowers suggests that it could play a regulatory role in cell division.

The involvement of *HMGB* genes in plant stress responses has been reported in several species, including *Arabidopsis* [22,45,46], rice [31], cucumber [15,32], *Solanum commersonii* [47], and *Betula platyphylla* [17]. For example, *MdHMGB15*, an ARID-HMG gene in apple, is upregulated under salt stress and activates the transcription of *MdXERICO*, which encodes a ubiquitin E3 ligase. This ligase promotes the ubiquitination and 26S proteasome-dependent degradation of MdNRP, thereby enhancing salt tolerance [48]. Similarly, the watermelon ARID-HMG genes *ClHMGB4* and *ClHMGB5* were upregulated under salt stress (Figure 6e,f), indicating that they may play a conserved role in salt-stress adaptation. *AtHMGB15* expression in *Arabidopsis* is induced by low-temperature, salt, and UV stress conditions, with the strongest response observed under cold conditions [46]. In contrast, its watermelon homologs *ClHMGB4* and *ClHMGB5* were upregulated by salt stress but downregulated under low-temperature. In addition, *AtHMGB1* expression is insensitive to drought and low-temperature but downregulated under high salinity, whereas *AtHMGB2* and *AtHMGB3* are upregulated by cold stress and downregulated under drought and high salinity [22]. Their respective watermelon orthologs—*ClHMGB3*, *ClHMGB9*, and *ClHMGB7* (Table 1)—show both conserved and divergent expression profiles. In watermelon, *ClHMGB3* was downregulated under drought and upregulated under low-temperature (Figure 6a,c), while *ClHMGB7* and *ClHMGB9* were also downregulated under drought and upregulated after 24 h of low-temperature (Figure 6a,c). The delayed cold response of *ClHMGB7* and *ClHMGB9* may reflect species-specific differences in cold sensitivity. Moreover, contrasting expression patterns between *ClHMGB3* and *AtHMGB1*, and between *ClHMGB9* and *AtHMGB2* under salt stress suggest that species-specific regulatory mechanisms modulate *HMGB* gene function in stress adaptation. Collectively, the varied expression profiles of homologous *HMGB* genes under different stresses point to distinct regulatory networks that have evolved in different plant lineages.

Research on the involvement of *HMGB* genes in plant biotic stress responses remains relatively limited. To date, *Arabidopsis HMGB3* has been shown to activate innate immune responses against *Botrytis cinerea* infection [36], while rice *OsHMGB1* negatively regulates resistance to *Xoo* but positively modulates defense against root-knot nematodes [37,38]. In this study, we used the highly destructive fungal pathogen *Fon*—the causal agent of Fusarium wilt in watermelon—to examine the expression patterns of *ClHMGB* genes under biotic stress. In the *Fon* R2-susceptible cultivar ‘M08’ [49,50], more *ClHMGB* genes were upregulated at all three sampling time points after inoculation (Figure 7a). Interestingly, in a previous study on watermelon *SWEET* genes, we observed that infection with *Fon* R2 resulted in a notably higher number of upregulated *ClaSWEET* genes in susceptible cultivars compared to resistant ones [51]. A similar trend has been reported in cucumber, where approximately half of the chitinase gene family members were upregulated upon *Fusarium oxysporum* infection in susceptible varieties, compared to only about one-quarter in resistant varieties [52]. Together, these findings suggest that plants may activate the expression of a wider array of genes in response to invasion by highly virulent pathogens. Notably, all *ClHMGB* genes induced by *Fon* R1 were also upregulated under *Fon* R2 infection (Figure 7a). Among them, *ClHMGB1*, *4*, *6*, and *7* showed sustained upregulation during infection by both races (Figure 7b), indicating that they may play a core role in watermelon resistance to Fusarium wilt. Future studies should aim to clarify the specific functions of these four genes in mediating the watermelon–*Fon* interaction.

This study involves a comprehensive analysis of the *ClHMGB* gene family in watermelon. The identification of its members, along with insights into their expression patterns under various stresses, establishes a foundation for elucidating the biological functions of ClHMGB proteins in watermelon development and stress responses. The distinct expression profiles of certain genes (e.g., *ClHMGB4* in flowers; *ClHMGB1*, *4*, *6*, and *7* during Fusarium wilt infection) suggest their potential regulatory roles in development and stress adaptation, offering new entry points for subsequent functional studies. However, the precise mechanisms through which *ClHMGB* members mediate responses to different abiotic stresses and confer resistance to Fusarium wilt remain unknown. Future research employing genetic knockdown/overexpression, protein interaction assays, and detailed phenotyping under stress conditions is needed to establish causal relationships. Clarifying these roles could inform targeted strategies for breeding crops with enhanced resilience to diseases and environmental stresses.

## 4. Materials and Methods

### 4.1. Plant Materials, as Well as Abiotic and Biotic Stress Treatments

The watermelon cultivar ‘M08’, provided by the Gourd Germplasm Resources Research Group at the College of Horticulture, Northwest A&F University, was used to analyze the expression patterns of *ClHMGB* genes under abiotic and biotic stress, and in various tissues. Seeds were surface-sterilized with 75% ethanol and sown in nursery substrate. Plants were grown in a controlled-climate chamber under the following conditions: 28 °C and a 14 h light/22 °C, 10 h dark photoperiod. *Nicotiana benthamiana* plants used for subcellular localization were grown in a controlled-climate chamber with the same conditions.

Abiotic stress treatments included drought, low-temperature, and salt stress. For biotic stress, two physiological races of *Fusarium oxysporum* f. sp. *niveum* (*Fon* R1 and *Fon* R2) were used. All stress treatments and sampling procedures followed the protocol described by Xuan [49].

### 4.2. Identification and Characterization of HMGB Genes in Watermelon

The BLAST search of the watermelon genome database was performed using TBtools software (Version 2.360) with the HMGB protein sequences of *Arabidopsis* as queries (E-value cutoff of 10^−10^; http://cucurbitgenomics.org/organism/21, accessed on 10 July 2025). The putative ClHMGB protein sequences were submitted to the Pfam database (http://pfam.xfam.org/search/sequence, accessed on 11 July 2025), SMART database (http://smart.embl-heidelberg.de, accessed on 12 July 2025), and NCBI conserved domain database (https://www.ncbi.nlm.nih.gov/cdd, accessed on 12 July 2025) to confirm the conserved domains. The lengths of the genomic sequences and coding sequences of the *ClHMGB* genes were obtained from the GFF genome annotation file. In addition, the subcellular localization of each protein was predicted using WoLF PSORT (http://www.genscript.com/psort/wolf_psort.html, accessed on 12 July 2025), while the physicochemical properties, including molecular weight and theoretical isoelectric point, were analyzed with the ProtParam tool (http://expasy.org, accessed on 16 July 2025).

### 4.3. Phylogenetic, Syntenic, Gene Structure, Conserved Domain, and Cis-Acting Regulatory Element Analyses

To examine the evolutionary relationships of *HMGB* genes across species, the corresponding HMGB protein sequences were retrieved from the *Arabidopsis* (*Arabidopsis thaliana*) database (https://www.arabidopsis.org/, accessed on 2 August 2025), the tomato (*Solanum lycopersicum*) genome in Ensembl Plants (https://plants.ensembl.org/Solanum_lycopersicum/Info/Index, accessed on 3 August 2025), and the Rice Gene Database (http://www.ricedata.cn/gene/, accessed on 5 August 2025). An unrooted phylogenetic tree was constructed from these sequences using MEGA 7.0.21 with the neighbor-joining method and 1000 bootstrap replicates, and the tree was subsequently refined using the iTOL online tool (https://itol.embl.de/, accessed on 5 August 2025). The syntenic analysis of watermelon, *Arabidopsis*, rice, and tomato was performed with the One-Step MCScanX program in TBtools (Version 2.360) using the default parameters, based on the genome and annotation files of the four species. The identified syntenic *HMGB* gene pairs were visualized with the Dual Synteny Plot tool for MCScanX. The chromosomal locations of *ClHMGB* genes were mapped using MG2C (http://mg2c.iask.in/mg2c_v2.1/index_cn.html, accessed on 17 August 2025), and their exon–intron structures were illustrated with the Gene Structure Display Server (https://gsds.gao-lab.org/, accessed on 20 August 2025). Multiple sequence alignment of HMGB proteins from the four species was performed using Clustal X (Version 2.0), with similar amino acid sequences highlighted. Subsequently, the ClHMGB protein sequences were submitted to the Multiple Em for Motif Elicitation database (https://meme-suite.org/meme/, accessed on 25 August 2025) for conserved motif analysis. To identify *cis*-acting regulatory elements, 2 kb sequences upstream of the start codon of each *ClHMGB* gene were extracted and analyzed with the PlantCARE database (https://bioinformatics.psb.ugent.be/webtools/plantcare/html/, accessed on 27 August 2025).

### 4.4. RNA Extraction and Expression Pattern Analysis

Total RNA was extracted from the samples using the RNAsimple Total RNA Kit (TianGen, Beijing, China) following the manufacturer’s protocol. First-strand cDNA was synthesized from approximately 1 μg of total RNA with the FastKing RT Kit (TianGen, Beijing, China). Quantitative reverse transcription PCR (qRT-PCR) was carried out in a 20 μL reaction mixture containing 10 μL of SYBR^®^ Green I Master Mix (Aidlab, Beijing, China), 0.8 μL each of forward and reverse primers, 2 μL of cDNA template, and 6.4 μL of ddH_2_O. Gene-specific primers for the target *ClHMGB* genes and the reference gene *Claactin-7* were used for amplification (Appendix A). All reactions were performed with three biological replicates. Relative expression levels were calculated using the 2^−ΔΔCT^ method. Expression values were log2-transformed and normalized, and the heatmaps were generated using MeV 4.8.1 software.

### 4.5. Subcellular Localization

The cDNA sequences of *ClHMGB1* and *ClHMGB2* were cloned and inserted into the pGreen-GFP vector and the resulting recombinant plasmids were introduced into *Agrobacterium* strain GV3101 cells via heat-shock transformation. Healthy *Nicotiana benthamiana* plants were selected for agroinfiltration, wherein the abaxial side of leaves was injected with *Agrobacterium* suspensions carrying the fusion constructs [53]. After 36 h of dark cultivation, subcellular localization of the GFP-tagged proteins was examined by confocal laser scanning microscopy. To confirm nuclear localization, the *Arabidopsis* histone H2B (AtH2B) protein fused with RFP was co-expressed as a nuclear marker [54].

## 5. Conclusions

This study systematically identified nine *HMGB* genes in watermelon using a genome-wide search, dividing them into four classes. Synteny analysis revealed a closer phylogenetic relationship of *ClHMGB* genes with dicot species, and the analyses of *cis*-acting elements and expression patterns suggest that *ClHMGB* genes may play roles in tissue development and in responses to various biotic and abiotic stresses. Nevertheless, transcript-level changes alone cannot determine their regulatory direction; for instance, while *ClHMGB4* exhibited upregulation in female flowers, as well as under salt stress and during Fusarium wilt infection, it remains unclear whether it plays a positive regulatory role in these processes. Future research will employ molecular genetic approaches to functionally characterize these candidate genes and elucidate their precise mechanisms in the development and environmental stress responses of watermelon.

## Figures and Tables

**Figure 1 ijms-27-00157-f001:**
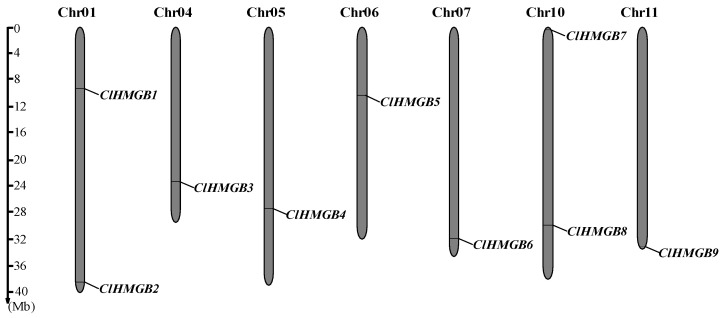
Schematic representation of the genomic locations of the nine *ClHMGB* genes across the seven watermelon chromosomes. The relative positions of the *ClHMGBs* are marked on the chromosomes. The scale bar on the left side shows the physical distance (Mb).

**Figure 2 ijms-27-00157-f002:**
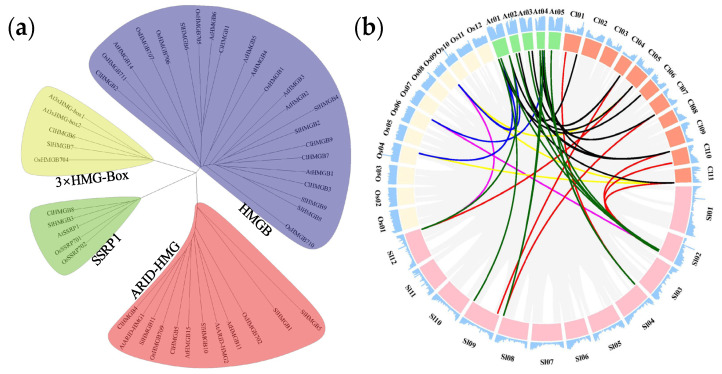
Phylogenetic and syntenic analysis of *HMGB* genes. (**a**) An unrooted phylogenetic tree was constructed in MEGA 7.0.21 from the amino acid sequences of HMGB proteins from *Arabidopsis*, rice, tomato, and watermelon using the neighbor-joining method with 1000 bootstrap interactions. The tree delineates four major clades, color-coded as follows: HMGB (purple), ARID-HMG (red), SSRP1 (green), and 3xHMG-box (yellow). (**b**) Syntenic relationships of *HMGB* genes among *Arabidopsis*, rice, tomato, and watermelon were analyzed using the One-Step MCScanX program in TBtools (Version 2.360) with the default parameters. The colored lines connecting different genomes indicate different syntenic *HMGB* gene pairs: *Arabidopsis* and watermelon (black), tomato and watermelon (red), rice and watermelon (yellow), *Arabidopsis* and tomato (green), *Arabidopsis* and rice (blue), and rice and tomato (purple).

**Figure 3 ijms-27-00157-f003:**
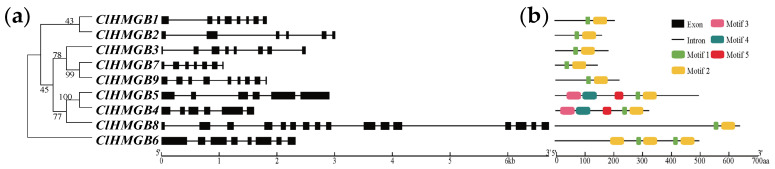
Phylogenetic relationships and structures of *ClHMGB* genes and conserved motifs of ClHMGB. (**a**) Phylogenetic tree and exon–intron organization of *ClHMGB* family members. Exons and introns are depicted by black boxes and lines, respectively. (**b**) Distribution of conserved protein motifs identified in ClHMGB proteins, with distinct colored squares representing five unique motif sequences. The two scale bars below the image indicate the physical lengths of the gene (kb) and the protein (aa), respectively.

**Figure 4 ijms-27-00157-f004:**
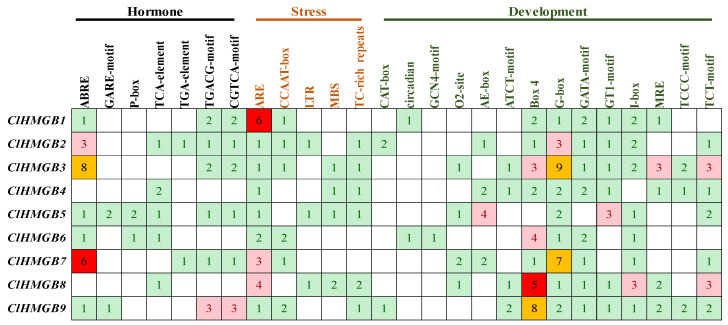
Analysis of *cis*-acting elements in the promoters of *ClHMGB* genes. The numbers illustrate the frequency of *cis*-elements associated with phytohormone responsiveness, stress responsiveness, and plant growth and development. The colors represent distinct frequency ranges: green (1–2), light red (3–4), dark red (5–6), and yellow (>7). Phytohormone responsiveness: ABRE (abscisic acid); GARE-motif and P-box (gibberellin); TCA-element (salicylic acid); TGA-element (auxin); TGACG-motif; and CGTCA-motif (MeJA). Stress responsiveness: ARE (anaerobic induction); CCAAT-box (heat stress); LTR (low-temperature); MBS and TC-rich repeats (defense/stress). Plant growth and development: CAT-box (meristem); circadian (circadian control); GCN4_motif (endosperm); O2-site (zein metabolism); and light-responsive elements (AE-box, ATCT-motif, Box4, G-box, GATA-motif, GT1-motif, I-box, MRE, TCCC-motif, and TCT-motif).

**Figure 5 ijms-27-00157-f005:**
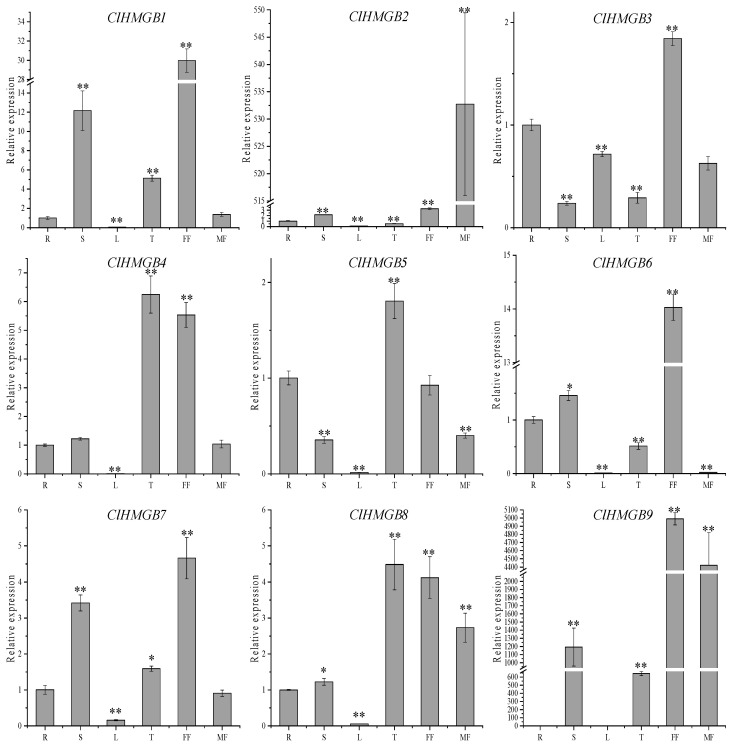
Expression analysis of *ClHMGB* genes in various watermelon tissues. The relative expression level in each tissue (stem (S), leaf (L), tendril (T), female flower (FF), and male flower (MF)) was calculated by setting the expression in the root (R) to 1. Data were normalized to the expression of *Claactin-7*. Values are presented as means ± SE (*n* = 3). Significant differences relative to the root are indicated (* *p* < 0.05 and ** *p* < 0.01 according to Student’s *t*-test).

**Figure 6 ijms-27-00157-f006:**
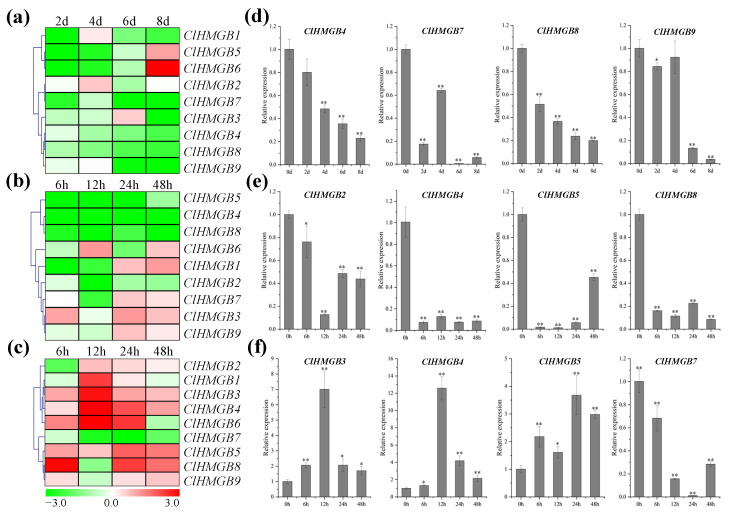
Expression profiling of *ClHMGB* genes under abiotic stresses. Heatmaps depicting transcript abundance under drought (**a**), low-temperature (**b**), and salt (**c**) stress conditions. Red and green correspond to strong (log2FC > 0.0) and weak (log2FC < 0.0) expression of the *ClHMGB* genes, respectively. The phylogenetic tree was built using the average linkage clustering method. The plant samples at 0 dpt and 0 hpt were considered to be controls (log2FC = 0.0). (**d**–**f**) show detailed expression patterns of the *ClHMGB*s in response to drought, low-temperature, and salt stress, respectively. dpt: days post-treatment; hpt: hours post-treatment. For each gene, expression at 0 dpt/0 hpt was set to 1 to calculate the relative expression at subsequent time points. Data were normalized to *Claactin-7*, and are presented as means ± SE (*n* = 3). Asterisks indicate significant differences compared to the 0 dpt/0 hpt control (* *p* < 0.05 and ** *p* < 0.01 according to Student’s *t*-test).

**Figure 7 ijms-27-00157-f007:**
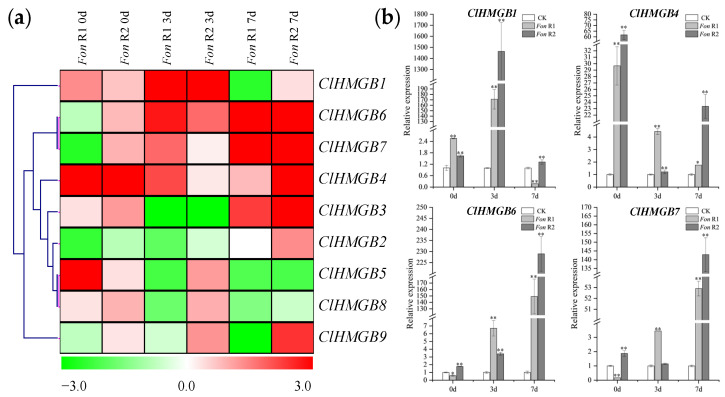
Expression patterns of *ClHMGB* genes in response to *Fon* infection. (**a**) Heatmap of the expressed *ClHMGB* genes in the root tissue of ‘M08’ after infection with the causal agent of Fusarium wilt. The gene clusters were generated using the average linkage clustering method. Red and green correspond to strong (log2FC > 0.0) and weak (log2FC < 0.0) expression of the *ClHMGB* genes, respectively. The plant samples at 0 dpt were considered to be controls (log2FC = 0.0). *Fon* R1, *Fusarium oxysporum* f. sp. *niveum* race 1; *Fon* R2, *Fusarium oxysporum* f. sp. *niveum* race 2. (**b**) Detailed expression patterns of *ClHMGB1*, *4*, *6*, and *7* infected with *Fon* R1 and *Fon* R2 at 0, 3, and 7 dpt. For each gene, the expression level of the untreated control at each time point was set to 1, and the relative expression levels upon inoculation with either *Fon* R1 or *Fon* R2 were calculated accordingly. Data were normalized to *Claactin-7*, and are presented as means ± SE (*n* = 3). Asterisks indicate significant differences compared to the 0 dpt control (* *p* < 0.05 and ** *p* < 0.01 according to Student’s *t*-test).

**Table 1 ijms-27-00157-t001:** Characteristics of the *ClHMGB* genes in watermelon.

Gene Name	Gene ID	*Arabidopsis* *Ortholog Locus*	E-Value	Genomic Sequence (bp)	CDS(bp) ^1^	Protein Length (aa)	MW (kDa) ^2^	pI ^3^	Subcellular Localization
*ClHMGB1*	Cla97C01G008330.1	AT5G23420.1/*AtHMGB6*	2.0 × 10^−44^	1822 (−)	624	208	22.91	6.69	N
*ClHMGB2*	Cla97C01G024440.1	AT2G34450.1/*AtHMGB14*	1.0 × 10^−39^	3011 (−)	489	163	18.94	9.39	N
*ClHMGB3*	Cla97C04G073670.1	AT3G51880.4/*AtHMGB1*	2 × 10^−26^	2496 (−)	552	184	20.78	6.07	N
*ClHMGB4*	Cla97C05G096410.1	AT1G76110.1/*AtARID-HMG1*	1.0 × 10^−123^	1604 (+)	975	325	37.37	9.55	N
*ClHMGB5*	Cla97C06G117510.1	AT1G04880.1/*AtHMGB15*	1.0 × 10^−132^	2909 (+)	1497	499	55.92	4.88	N
*ClHMGB6*	Cla97C07G141850.1	AT4G23800.2/*At3xHMG-box2*	1.0 × 10^−113^	2324 (+)	1503	501	58.6	9.19	N
*ClHMGB7*	Cla97C10G184790.1	AT1G20696.2/*AtHMGB3*	1.0 × 10^−32^	1073 (+)	441	147	16.11	8.51	N
*ClHMGB8*	Cla97C10G197730.1	AT3G28730.1/*AtSSRP1*	0	6714 (−)	1929	643	71.61	5.73	N
*ClHMGB9*	Cla97C11G224900.1	AT1G20693.3/*AtHMGB2*	2.0 × 10^−44^	1823 (+)	663	221	23.93	9.08	N

^1^ Length of coding sequence (CDS); ^2^ molecular weight (MW); ^3^ theoretical isoelectric point (pI); N, nucleus.

## Data Availability

The original contributions presented in this study are included in the article/Appendix A; further inquiries can be directed to the corresponding author.

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
