# Peer review of "Genome-Wide Identification and Expression Analysis of the ClHMGB Gene Family in Watermelon Under Abiotic Stress and Fusarium oxysporum Infection"

_ijms, 2025, doi:10.3390/ijms27010157_

Round 1
Reviewer 1 Report
Comments and Suggestions for Authors
The manuscript is titled “Genome-wide identification and expression analysis of the ClHMGB gene family in watermelon under abiotic stress and Fusarium oxysporum infection” The content of the original manuscript has a certain theoretical significance, but the English writing mistakes need to be carefully polished, and the logical relationship must be strengthened. However, there are some points that need to be addressed before publication.
The following problems were identified.
- Please note that the entire text should be written in the format of genes and proteins and requires full manuscript proofreading. When referring to genes, they should be written in italics.
- In the Results section, Figures 5, 6, and 7 did not undergo significance analysis, compromising the stability of the data.
- In the Results section, the pixel resolution of all images needs to be improved, and the captions need to be detailed so that readers can clearly understand what the images are conveying. All images require an improved layout to enhance readability; this issue necessitates modifications to all images throughout the manuscript.
- In the Discussion section, it is necessary to explore whether this research demonstrates novelty, the scientific questions it addresses, and its potential for advancing this field.
- The conclusion should concisely summarize the key focus and central themes of the study, highlighting its findings and limitations, as well as future research plans. This requires reworking and drafting.
- The references are excessively outdated, with few publications in the past three years. Additionally, there is a lack of volume and issue numbers, and page references. The reference format requires standardization and revision.
- This manuscript requires a more logically structured organization, with the English sections also needing corresponding adjustments in terms of specialized terminology, grammar, and coherence.
Author Response
|
Comments 1: [Please note that the entire text should be written in the format of genes and proteins and requires full manuscript proofreading. When referring to genes, they should be written in italics.] |
|
Response 1: Thank you for pointing this out. We agree with this comment. We sincerely apologize for the inconsistency in the formatting (italic vs. non-italic) of gene and protein names caused by document conversion between different versions. This error has been corrected in the latest version of the manuscript. |
|
Comments 2: [In the Results section, Figures 5, 6, and 7 did not undergo significance analysis, compromising the stability of the data.] |
|
Response 2: Agree. Thank you for your valuable suggestion. Accordingly, we have performed statistical significance analysis on the data in Figures 5, 6, and 7 of the latest revised manuscript. |
|
Comments 3: [In the Results section, the pixel resolution of all images needs to be improved, and the captions need to be detailed so that readers can clearly understand what the images are conveying. All images require an improved layout to enhance readability; this issue necessitates modifications to all images throughout the manuscript.] |
|
Response 3: Agree. In response to this suggestion, we have taken the following actions: the resolution of all figures in the Results section has been increased, the figure captions have been elaborated, and the layout of the figures has been optimized. These revisions are intended to ensure that readers can clearly understand the information presented in each figure and to enhance the overall readability of the manuscript. |
|
Comments 4: [In the Discussion section, it is necessary to explore whether this research demonstrates novelty, the scientific questions it addresses, and its potential for advancing this field.] |
|
Response 4: Thank you for this insightful suggestion. We have thoroughly revised the Discussion section to explicitly address the points you raised. Specifically, we added content (lines 327–336) to highlight the novel contributions of our work and revised the concluding paragraph (lines 425–437) to clarify the scientific questions addressed and the potential implications of this research for advancing the field. |
|
Comments 5: [The conclusion should concisely summarize the key focus and central themes of the study, highlighting its findings and limitations, as well as future research plans. This requires reworking and drafting.] |
|
Response 5: Agree. Thank you for this constructive suggestion. We have fully revised the conclusion section to provide a clearer and more concise summary of the study's key findings, limitations, and future research plans. The updated text is now included in the revised manuscript. Line 514-524. |
|
Comments 6: [The references are excessively outdated, with few publications in the past three years. Additionally, there is a lack of volume and issue numbers, and page references. The reference format requires standardization and revision.] |
|
Response 6: Agree. In the latest version of the manuscript, we have updated outdated references, supplemented missing volume, issue, and page numbers, and standardized the formatting of all references throughout the manuscript. |
|
Comments 7: [This manuscript requires a more logically structured organization, with the English sections also needing corresponding adjustments in terms of specialized terminology, grammar, and coherence.] |
|
Response 7: Agree. Thank you for your suggestion. We acknowledge the presence of some logical, terminological, and grammatical issues in our manuscript. The manuscript has undergone professional language polishing by a native editor and has been further reviewed for scientific logic and content accuracy by a subject-matter expert in biology and genomics. |

Reviewer 2 Report
Comments and Suggestions for Authors
It is an interesting manuscript. The authors present a comprehensive genome-wide identification and characterization of the HMGB gene family in watermelon. The classification and comparative genomics, including synteny and motif analysis across four species, are clearly presented. The expression experiments covering multiple abiotic stresses and two pathogen races are biologically relevant. The aims of this study are clear. I do not have major comments, but I have several suggestions for improving the manuscript.
Lines 211–216: Promoter elements such as ABRE, DRE, and light-responsive motifs are common in plant promoters. The authors use Figure 4 to present information on cis-acting elements in the ClHMGB promoters. However, I could not find the corresponding description of this figure in the main text. In addition, the interpretation should be cautious and should not imply direct functional regulation.
Lines 275–350: The authors use descriptive terms such as “upregulated” and “downregulated,” but these remain qualitative rather than quantitative. Although color gradients are shown in Figures 6 and 7, it is still necessary to provide fold-change thresholds (e.g., ≥2-fold) or include additional analyses such as hierarchical clustering or heatmaps.
Lines 335–345, 476–481: The manuscript proposes roles in floral development, abiotic stress tolerance, and defense against Fusarium, but these statements are based solely on expression patterns. Although the results show correlations between expression profiles and phenotypes, there is still no direct evidence such as knockout, complementation, or overexpression assays. Without functional validation, these claims should be rephrased using terms such as “potential” or “putative.”
Lines 379–384: The authors state that Fusarium race 2 induces more genes. This is interesting, but it remains unclear whether differences in infection severity, timing, or pathogen load contributed to the observed patterns. More context and discussion are needed.
Figures 5–7: I understand that the bars are visually separated; however, statistical tests are still required to support whether the differences are significant. In the main text, the authors use the term “significantly” to describe results shown in Figures 5–7. The authors should avoid using the word “significantly” unless the claim is supported by statistical data.
Comments on the Quality of English Language
The manuscript is understandable and mostly clear, but the English contains numerous issues that would require professional language polishing before publication.
Line 15–16: please revise “…we identified nine ClHMGB genes in watermelon (Citrullus lanatus [Thunb.]) through genome-wide screening.” to ”…we identified nine ClHMGB genes in watermelon using a genome-wide search.”
Line 27–28: please revise “…suggesting their core functions in watermelon disease resistance.” to ”…suggesting that they may play fundamental roles in watermelon disease resistance.”
Line 36–37: please revise ”They are named for their characteristically rapid migration during gel electrophoresis, which results from their relatively small size compared to other chromatin proteins.” to “They are named for their rapid mobility during gel electrophoresis, a consequence of their relatively small size.”
Line 57–59: please revise “As previously reported, HMGB proteins are involved in a wide range of critical biological functions, including the regulation of DNA replication, transcription, recombination, and repair, furthermore, they participate in plant growth, and development.” to “As previously reported, HMGB proteins are involved in a wide range of critical biological functions, including the regulation of DNA replication, transcription, recombination, and repair. Furthermore, they participate in plant growth and development.”
Line 72–74: please revise “Furthermore, it was revealed that OsHMGB1 maintains Pi homeostasis by binding to the promoters of a set of phosphate starvation response (PSR) genes to regulate their expression” to “Furthermore, OsHMGB1 was shown to maintain Pi homeostasis by binding to the promoters of phosphate starvation response (PSR) genes and regulating their expression.”
Line 138–139: please revise “…using the homologous HMGB protein sequences from Arabidopsis as queries.” to “…using HMGB protein sequences from Arabidopsis as queries.”
Line 143–146: please revise “According to the genomic data, these ClHMGB genes exhibited genomic sequence lengths ranging from 1073 to 6714 bp, coding sequence (CDS) lengths from 441 to 1929 bp, encoded amino acid lengths from 147 to 643 aa, and molecular weights ranging from 16.11 to 71.61 kDa.” to “Based on genomic data, the ClHMGB genes have genomic sequence lengths ranging from 1073 to 6714 bp, coding sequence (CDS) lengths from 441 to 1929 bp, protein lengths from 147 to 643 amino acids, and molecular weights ranging from 16.11 to 71.61 kDa.”
Line 336–338: please revise “…suggesting that this gene family may play a key regulatory role in watermelon flower organ development.” to “…suggesting that this gene family may play key regulatory roles in floral organ development.”
Line 365-367: please revise “Similarly, ClHMGB7 and ClHMGB9 are down-regulated under drought and up-regulated after 24 hours of low-temperature treatment Figure 6a and 6c).” to “Similarly, ClHMGB7 and ClHMGB9 are downregulated under drought and are upregulated after 24 hours of low-temperature treatment (Figure 6a and 6c).”
Line 413: please revise “The BLAST search was perform in the watermelon genome database…” to “The BLAST search was performed in the watermelon genome database…”
Author Response
|
Comments 1: [Lines 211–216: Promoter elements such as ABRE, DRE, and light-responsive motifs are common in plant promoters. The authors use Figure 4 to present information on cis-acting elements in the ClHMGB promoters. However, I could not find the corresponding description of this figure in the main text. In addition, the interpretation should be cautious and should not imply direct functional regulation.] |
|
Response 1: Thank you for your suggestion. We sincerely apologize that the description of Figure 4 was inadvertently omitted from the previous version due to file format conversion issues between documents. This content has now been restored in the latest revised manuscript, spanning lines 219 to 240. |
|
Comments 2: [Lines 275–350: The authors use descriptive terms such as “upregulated” and “downregulated,” but these remain qualitative rather than quantitative. Although color gradients are shown in Figures 6 and 7, it is still necessary to provide fold-change thresholds (e.g., ≥2-fold) or include additional analyses such as hierarchical clustering or heatmaps.] |
|
Response 2: Agree. Thank you for pointing out this limitation in our manuscript. In the revised version, we have provided more detailed explanations in the captions of Figures 6 and 7, specified the fold‑change threshold, and added statistical significance analysis to the histogram data. |
|
Comments 3: [Lines 335–345, 476–481: The manuscript proposes roles in floral development, abiotic stress tolerance, and defense against Fusarium, but these statements are based solely on expression patterns. Although the results show correlations between expression profiles and phenotypes, there is still no direct evidence such as knockout, complementation, or overexpression assays. Without functional validation, these claims should be rephrased using terms such as “potential” or “putative.”] |
|
Response 3: Yes, thank you very much for this valuable suggestion. In the revised manuscript, we have carefully reviewed the relevant wording and used terms such as "potential" or "putative" to describe the possible functions of these genes. |
|
Comments 4: [Lines 379–384: The authors state that Fusarium race 2 induces more genes. This is interesting, but it remains unclear whether differences in infection severity, timing, or pathogen load contributed to the observed patterns. More context and discussion are needed.] |
|
Response 4: Agree. Thank you for pointing out this limitation in our manuscript. In the revised version, we have added two previously reported similar cases and included additional discussion to better support our observations. Now, line 412-419. |
|
Comments 5: [Figures 5–7: I understand that the bars are visually separated; however, statistical tests are still required to support whether the differences are significant. In the main text, the authors use the term “significantly” to describe results shown in Figures 5–7. The authors should avoid using the word “significantly” unless the claim is supported by statistical data.] |
|
Response 5: Agree. Thank you for your valuable suggestions. In the revised manuscript, we have modified the descriptive wording accordingly and performed statistical tests on the data presented in Figures 5–7 to confirm the significance of the observed differences. |
|
4. Response to Comments on the Quality of English Language |
|
Point 1: Line 15–16: please revise “…we identified nine ClHMGB genes in watermelon (Citrullus lanatus [Thunb.]) through genome-wide screening.” to ”we identified nine ClHMGB genes in watermelon using a genome-wide search.” |
|
Response 1: Yes. We have corrected this part to” we identified nine ClHMGB genes in watermelon using a genome-wide search.” Now, line15. |
|
Point 2: Line 27–28: please revise “suggesting their core functions in watermelon disease resistance.” to ”suggesting that they may play fundamental roles in watermelon disease resistance.” |
|
Response 2: Yes. We have corrected this part to” suggesting that they may play fundamental roles in watermelon disease resistance.” Now, line 26-27. |
|
Point 3: Line 36–37: please revise ”They are named for their characteristically rapid migration during gel electrophoresis, which results from their relatively small size compared to other chromatin proteins.” to “They are named for their rapid mobility during gel electrophoresis, a consequence of their relatively small size.” |
|
Response 3: Yes. We have corrected this part to” They are named for their rapid mobility during gel electrophoresis, a consequence of their relatively small size.” Now, line 35-36. |
|
Point 4: Line 57–59: please revise “As previously reported, HMGB proteins are involved in a wide range of critical biological functions, including the regulation of DNA replication, transcription, recombination, and repair, furthermore, they participate in plant growth, and development.” to “As previously reported, HMGB proteins are involved in a wide range of critical biological functions, including the regulation of DNA replication, transcription, recombination, and repair. Furthermore, they participate in plant growth and development.” |
|
Response 4: Yes. We have corrected this part to” As reported previously, HMGB proteins are involved in a wide range of critical biological processes, such as the regulation of DNA replication, transcription, recombination, and repair. And also participate in plant growth and development.’’ Now, line 54-57. |
|
Point 5: Line 72–74: please revise “Furthermore, it was revealed that OsHMGB1 maintains Pi homeostasis by binding to the promoters of a set of phosphate starvation response (PSR) genes to regulate their expression” to “Furthermore, OsHMGB1 was shown to maintain Pi homeostasis by binding to the promoters of phosphate starvation response (PSR) genes and regulating their expression.” |
|
Response 5: Yes. We have corrected this part to” Further studies revealed that OsHMGB1 maintains Pi homeostasis by binding to promoters of phosphate starvation response (PSR) genes and regulating their expression’’ Now, line 69-71. |
|
Point 6: Line 138–139: please revise “…using the homologous HMGB protein sequences from Arabidopsis as queries.” to “…using HMGB protein sequences from Arabidopsis as queries.” |
|
Response 6: Yes. We have corrected this part to” using HMGB protein sequences from Arabidopsis as queries.’’ Now, line 131. |
|
Point 7: Line 143–146: please revise “According to the genomic data, these ClHMGB genes exhibited genomic sequence lengths ranging from 1073 to 6714 bp, coding sequence (CDS) lengths from 441 to 1929 bp, encoded amino acid lengths from 147 to 643 aa, and molecular weights ranging from 16.11 to 71.61 kDa.” to “Based on genomic data, the ClHMGB genes have genomic sequence lengths ranging from 1073 to 6714 bp, coding sequence (CDS) lengths from 441 to 1929 bp, protein lengths from 147 to 643 amino acids, and molecular weights ranging from 16.11 to 71.61 kDa.” |
|
Response 7: Yes. We have corrected this part to” Based on genomic data, the ClHMGB genes have genomic sequence lengths ranging from 1073 to 6714 bp, coding sequence (CDS) lengths from 441 to 1929 bp, protein lengths from 147 to 643 amino acids, and molecular weights ranging from 16.11 to 71.61 kDa.’’ Now, line 135-137. |
|
Point 8: Line 336–338: please revise “…suggesting that this gene family may play a key regulatory role in watermelon flower organ development.” to “…suggesting that this gene family may play key regulatory roles in floral organ development.” |
|
Response 8: Yes. We have corrected this part to” suggesting that this gene family may play key regulatory roles in floral organ development.’’ Now, line 369-370. |
|
Point 9: Line 365-367: please revise “Similarly, ClHMGB7 and ClHMGB9 are down-regulated under drought and up-regulated after 24 hours of low-temperature treatment Figure 6a and 6c).” to “Similarly, ClHMGB7 and ClHMGB9 are downregulated under drought and are upregulated after 24 hours of low-temperature treatment (Figure 6a and 6c).” |
|
Response 9: Yes. We have corrected this part to” while ClHMGB7 and ClHMGB9 were also downregulated under drought and upregulated after 24 hours of low-temperature treatment (Figure 6a and 6c).’’ Now, line 396-398. |
|
Point 10: Line 413: please revise “The BLAST search was perform in the watermelon genome database…” to “The BLAST search was performed in the watermelon genome database…” |
|
Response 10: Yes. We have corrected this part to” The BLAST search was performed in the watermelon genome database” Now, line 453. |

Reviewer 3 Report
Comments and Suggestions for Authors
Thank you for sending me this manuscript for revision. In this research, Changqing Xuan et al. using genome wide analysis found nine genes encoding High mobility Group B (HMGB) proteins from Citrus lanatus and were classified into four subfamilies by using phylogenic and homology analyses. Author performed tissue-specific expression profiling, finding that eight of these HMGB increased expression levels in female and male flowers concluding that these genes play essential roles in organ development. Author also analyzed stresses such low temperature, drought and salinity and two strains of Fusarium oxysporum where these HMGB displayed distinct expression patterns. Suggesting that these genes are involved in biotic and abiotic stress tolerance.
Overall, there are minor and major issues enlisted below.
1) In Figure 5. The data presented in ClHMGB9 and ClHMGB3 plots seem to be super high. how these genes would be expressed about 500 and 5000x times than the gene reference? Please explain.
2) Figure 6. It seems that the legend depiction is wrongly assigned (heatmaps should be a, b,c and plots d,e,f).
3) In Figure 6: Author displayed detailed expression patterns for certain genes based of the response to each stress. What kind of response positive or negative? As most of the genes are downregulated except CIHMGB3, CIHMGB4, and ClHMGB5.
4) Continuing with Figure 6, why do the authors not include a detailed expression pattern plot for missing ClHMGB missing (1, 6) ?
5) In figure 7 authors present plots for ClHMGB expression levels to evaluate the response to two Fusarium strains. One of my concern is the really high expression level relative to the control ranging up to 1400x increment which contrast to other plant species.
6) I would like to see the agarose gels showing RNA integrity or agarose gels for the gen control amplification versus the ClHMGB genes with upregulation pattern.
Author Response
|
3. Point-by-point response to Comments and Suggestions for Authors |
|
Comments 1: [In Figure 5. The data presented in ClHMGB9 and ClHMGB3 plots seem to be super high. how these genes would be expressed about 500 and 5000x times than the gene reference? Please explain.] |
|
Response 1: Thank you for pointing this out. We believe the genes you referred to should be ClHMGB2 and ClHMGB9 in Figure 5. In the tissue-specific expression analysis experiment, the relative expression level of each gene in stem, leaf, tendril, female flower, and male flower tissues was calculated by setting the expression level in root to 1. The reason you observed super high expression levels of ClHMGB2 and ClHMGB9 in other tissues is due to their very low expression levels in root tissue. Additionally, we have reviewed the raw data. In root tissue, the CT value difference between the ClHMGB2 primer and the Claactin-7 primer was 5.98, whereas in male flower tissue, it was -3.07. Similarly, for ClHMGB9, the CT value difference in root tissue was 11.86, while in tissues such as tendrils, it was 2.54 or lower. According to the calculation formula, the results indeed indicate that ClHMGB2 and ClHMGB9 exhibit particularly high expression levels in tissues such as male flowers. |
|
Comments 2: [Figure 6. It seems that the legend depiction is wrongly assigned (heatmaps should be a, b,c and plots d,e,f).] |
|
Response 2: Agree. Thank you for your thorough review of our manuscript. We appreciate you identifying the error in the legend depiction for Figure 6, and we have corrected it in the revised manuscript. |
|
Comments 3: [In Figure 6: Author displayed detailed expression patterns for certain genes based of the response to each stress. What kind of response positive or negative? As most of the genes are downregulated except CIHMGB3, CIHMGB4, and ClHMGB5] |
|
Response 3: Thank you for raising this question. We sincerely apologize that we are currently unable to definitively determine whether these responses are positive or negative as we can only describe the expression trends of different ClHMGB genes under various stress conditions. For example, although ClHMGB4 shows a downregulated expression pattern under drought and low-temperature stress, but exhibits an upregulated pattern under salt stress, we cannot yet conclude whether ClHMGB4 acts as a negative regulator in watermelon seedlings' response to drought and low-temperature stress, or as a positive regulator in their response to salt stress. Currently, we have selected several ClHMGB members for functional studies. In future research, we will provide a detailed explanation of the roles these selected members play in regulating watermelon’s ability to resist different stress conditions. |
|
Comments 4: [Continuing with Figure 6, why do the authors not include a detailed expression pattern plot for missing ClHMGB missing (1, 6) ?] |
|
Response 4: Thank you for pointing this out. In Figure 6, we examined the expression trends of the nine ClHMGB genes in watermelon under three different stresses, with a focus on those members that showed consistently upregulated or downregulated expression under a specific stress. Since ClHMGB1 and ClHMGB6 did not exhibit consistently increasing or decreasing expression trends across the three stress conditions, their detailed expression patterns were not provided. |
|
Comments 5: [In figure 7 authors present plots for ClHMGB expression levels to evaluate the response to two Fusarium strains. One of my concern is the really high expression level relative to the control ranging up to 1400x increment which contrast to other plant species.] |
|
Response 5: Yes, we have also observed this phenomenon. In a previous study on watermelon chitinase in response to Fusarium wilt, we found that ClHMGB1 acts as a transcription factor regulating the transcriptional expression of chitinase upon Fusarium wilt infection, and it was significantly upregulated in watermelon plants after infection (that study is currently unpublished). Additionally, we have initiated functional studies on this gene through knockout and overexpression approaches. In subsequent research, we will provide a detailed explanation of the specific roles of the ClHMGB1 gene in the interaction between watermelon and Fusarium wilt. |
|
Comments 6: [I would like to see the agarose gels showing RNA integrity or agarose gels for the gen control amplification versus the ClHMGB genes with upregulation pattern.] |
|
Response 6: Thank you for your suggestion. We are pleased to provide the requested gel images, including those for RNA integrity assessment and agarose gels comparing the control amplification with the upregulated ClHMGB genes. Due to the limited revision timeframe, we focused our validation on a subset of samples. Specifically, we performed RNA integrity electrophoresis on selected samples from Figure 7a and conducted a semi-quantitative analysis of the ClHMGB1 gene expression pattern as presented in Figure 7b. The corresponding results are provided below. |
|
The RNA integrity electrophoresis |
|
The semi-quantitative analysis of the ClHMGB1 gene expression pattern |
|
4. Response to Comments on the Quality of English Language |
|
Point 1: The English is fine and does not require any improvement. |
|
Response 1: Thank you very much for your positive feedback on the language quality of our manuscript. Following your comments, all language and technical issues have been reviewed and revised by a professional native English speaker with expertise in biology and genomics. |

Round 2
Reviewer 1 Report
Comments and Suggestions for Authors
This version is an improvement, but issues with image quality and text alignment within images remain and require rectification.
Author Response
Thank you for your valuable comments. In response to your feedback, we have thoroughly revised the images in our manuscript:
-
Image Quality Enhancement: We have reprocessed and regenerated all images to ensure they meet high-resolution and publication standards.
-
Text Alignment Correction: We have checked and realigned all labels, captions, and textual elements within the images to ensure accurate alignment and improve readability.
-
Language Polishing: Additionally, the manuscript has undergone professional refinement and optimization through the journal's (IJMS) officially recommended "Language Editing and Figure Enhancement Services" to further enhance its overall quality.
We believe that the issues you raised regarding the images have been properly addressed through these revisions. Thank you once again for your thorough review and constructive suggestions.
Reviewer 3 Report
Comments and Suggestions for Authors
I want to thank the Authors for a clear response to each of my comments. I do not have further questions or concerns. So, I happy recommending this manuscript for publication in this Journal.
Author Response
Thank you very much for your positive feedback and for recommending our manuscript for publication. We sincerely appreciate your time and insightful comments throughout the review process.